# Ceftolozane-Tazobactam Combination Therapy Compared to Ceftolozane-Tazobactam Monotherapy for the Treatment of Severe Infections: A Systematic Review and Meta-Analysis

**DOI:** 10.3390/antibiotics10010079

**Published:** 2021-01-15

**Authors:** Marco Fiore, Antonio Corrente, Maria Caterina Pace, Aniello Alfieri, Vittorio Simeon, Mariachiara Ippolito, Antonino Giarratano, Andrea Cortegiani

**Affiliations:** 1Department of Women, Child and General and Specialized Surgery, University of Campania “Luigi Vanvitelli”, 80138 Naples, Italy; antonio.corrente.md@gmail.com (A.C.); caterina.pace@libero.it (M.C.P.); anielloalfieri@gmail.com (A.A.); 2Medical Statistics Unit, Department of Public, Clinical and Preventive Medicine, University of Campania “Luigi Vanvitelli”, 80138 Naples, Italy; vittoriosimeon@gmail.com; 3Department of Surgical, Oncological and Oral Science (Di.Chir.On.S.), University of Palermo, 90127 Palermo, Italy; ippolito.mariachiara@gmail.com (M.I.); antonino.giarratano@unipa.it (A.G.); andrea.cortegiani@unipa.it (A.C.); 4Department of Anaesthesiology, Intensive Care and Emergency, Policlinico Paolo Giaccone, 90127 Palermo, Italy

**Keywords:** *pseudomonas aeruginosa*, ESBLs, multidrug resistance, β-lactamase inhibitors, anti-infective agents, bacteremia, ceftolozane, sepsis, infection, systematic review, meta-analysis

## Abstract

Ceftolozane-tazobactam (C/T) is a combination of an advanced-generation cephalosporin (ceftolozane) with a β-lactamase inhibitor (tazobactam). It is approved for the treatment of complicated urinary-tract/intra-abdominal infections and hospital-acquired/ventilator-associated pneumonia. This systematic review and meta-analysis (registered prospectively on PROSPERO, no. CRD42019134099, on 20 January 2020) aimed to evaluate the effectiveness of C/T combination therapy compared to C/T monotherapy for the treatment of severe infections and to describe the prevalence of microorganisms in the included studies. We retrieved literature from PubMed, EMBASE, and CENTRAL, until 26 November 2020. Eligible studies were both randomised trials and nonrandomised studies with a control group, published in the English language and peer-reviewed journals. The primary outcome was all-cause mortality; secondary outcomes were (i) clinical improvement and (ii) microbiological cure. Eight nonrandomised studies were included in the qualitative synthesis: Seven retrospective cohort studies and one case-control study. The meta-analysis of the four studies evaluating all-cause mortality (in total 148 patients: 87 patients treated with C/T alone and 61 patients treated with C/T combination therapy) showed a significant reduction of mortality in patients receiving C/T combination therapy, OR: 0.31, 95% CI: 0.10–0.97, *p* = 0.045. Conversely, the meta-analysis of the studies evaluating clinical improvement and microbiological cure showed no differences in C/T combination therapy compared to C/T monotherapy. The most consistent data come from the analysis of the clinical improvement, *n* = 391 patients, OR: 0.97, 95% CI: 0.54–1.74, *p* = 0.909. In 238 of the 391 patients included (60.8%), C/T was used for the treatment of infections caused by Pseudomonas aeruginosa.

## 1. Introduction

Ceftolozane-tazobactam (C/T) is an advanced-generation cephalosporin combined with a β-lactamase inhibitor approved for the treatment of complicated urinary tract infections (including pyelonephritis), complicated intra-abdominal infections (in combination with metronidazole), and for hospital-acquired (HAP)/ventilator-associated pneumonia (VAP) [1,2].

C/T is active against a common Gram-negative pathogen including ESBL-producing *Enterobacteriaceae* and *Pseudomonas aeruginosa*. Importantly, C/T retained potency against many multidrug resistant (MDR) and extensively drug resistant (XDR) strains [3].

In vitro studies evaluating combination regimens containing C-T plus amikacin [4,5,6], colistin [4], Fosfomycin, and aztreonam [7] showed an overall reduction in bacterial burden against multi-drug-resistant Gram-negative bacteria, especially *Pseudomonas aeruginosa.*

In contrast to pre-clinical studies, that seem to be all in favor of C/T combination therapy, clinical studies show discrepancies in the results. Moreover, there are no randomized controlled trials (RCT) in the literature that can give a high level of evidence to the question. Systematic reviews and meta-analyses help establish evidence-based clinical practice and resolve contradictory research outcomes, especially in the absence of large, well done RCT.

The aim of this systematic review and meta-analysis was to evaluate if in the clinical studies the C/T combination therapy is a more effective therapeutic strategy than C/T alone in the treatment of difficult-to-treat Gram-negative infections.

## 2. Materials and Methods

We registered the protocol after a search of the primary electronic registries (Cochrane Database of Systematic Reviews, the JBI Database of Systematic Reviews, and Implementation Reports and PROSPERO), to exclude the existing systematic review on the same topic, in the International Prospective Register of Systematic Reviews: PROSPERO (No. CRD42019134099) on 20 January 2020. We conducted a systematic review according to PRISMA methodology.

### 2.1. Study Search

The search strategy was performed following the PICO method (Table 1). The databases of the search included MEDLINE via PubMed, EMBASE, and Cochrane Central Register of Controlled Trials (CENTRAL). The search was conducted using the keyword “ceftolozane”, from inception to 13 May 2020. Then, the search was re-run, updating the data collection definitively until 26 November 2020.

### 2.2. Study Selection

We removed the duplicate after the search, and we listed all the included studies, using a citation management software (Endnote VX9. Clarivate Analytics, Philadelphia, PA, USA). We included as eligible studies both randomized clinical trials (RCT) and non-randomized studies with a control group, published in peer-reviewed journals in the English language. No restriction on the time of publication was applied. Two authors (AC and MI) evaluated the eligible studies with an initial screening based on the title and abstract, independently. The above-mentioned authors followed with a full-text screening of the selected articles for final inclusion. A third author (MF) resolved any disagreement on study eligibility or data extraction. The full text of the selected citations was assessed in detail by two independent reviewers (AC and MI) that recorded the reasons for exclusion of full-text studies that do not meet the inclusion criteria and reported in the systematic review. In addition, a final check was conducted by a third one (MF). The results of each step of the planned search have been reported in full version in the final report and presented in a Preferred reporting items for systematic reviews and meta-analyses (PRISMA) flow diagram (Figure 1).

### 2.3. Definition and Outcome

For the purpose of this study, we defined C/T combination therapy as the combined use of C/T and other antibiotic/s, and C/T monotherapy as the use of C/T as a single antibiotic therapy. The primary outcome was all-cause mortality. The secondary outcomes were clinical improvement and microbiological cure, respectively. For the secondary outcomes, we used the definitions provided by the authors of the included studies. All the outcomes were evaluated in patients who had a diagnosis of infection with at least one pathogen confirmed by the laboratory with any methods.

### 2.4. Data Extraction and Quality Assessment

Two authors (AC, MF) extracted data from the included studies using the Cochrane Data collection form for intervention reviews for RCTs and non-RCTs, independently. Two authors (AC, MF) assessed the methodological quality of the included studies using the Newcastle-Ottawa assessment scale (NOS) [8].

### 2.5. Data Analysis

We performed a meta-analysis using a conservative approach with the random-effects estimates of odds ratio (OR) for each outcome, which allows for the variation of real effects across studies, taken as “main results”. We quantified heterogeneity using the I^2^ statistic, which describes the percentage of total variation across studies that was attributable to heterogeneity rather than to chance. I^2^ values of 25%, 50%, and 75% correspond to cut-off points for low, moderate, and high degrees of heterogeneity. Sensitivity analyses evaluated whether a single study markedly affected the results. We used STATA version 16.0 (College Station, TX, USA) for all the analyses.

## 3. Results

### 3.1. Study Selection and Characteristics

Overall, we retrieved 1706 papers: 518 on PubMed, 1146 on EMBASE, and 42 on CENTRAL, among which we removed 317 duplicates. One thousand three hundred eighty-nine titles were identified as potentially relevant and screened, as shown in the flowchart (Figure 1). We excluded 1142 papers after the screening of the title and abstract of these 1389 articles. The main causes of exclusion of the 1142 papers were due to the fact that the articles were in vitro studies (328 papers), reviews/systematic review/meta-analysis (320 papers), case report/series (119 papers), or abstract/conference proceedings (70 papers). The table that summarizes the reasons of exclusion of all the retrieved papers is in Appendix A.

Of the remaining 27 studies, we excluded 19 studies from the full-text evaluation for four main reasons (Figure 1).

In total, after the full-text screening, we included eight nonrandomised studies in the qualitative synthesis (Table 2). Of these eight studies, seven were retrospective cohort studies (two multicenter and five single center) and one single center case-control study. Only one of the two multicenter studies was transnational (Rodríguez-Núñez et al.), of the seven non-transnational studies two were from the USA (Haidar et al. and Gerlach et al.) and five European (three from Spain and two from Italy). Seven of the eight included studies that evaluated infections due to *Pseudomonas aeruginosa*. Only Bassetti et al. evaluated the infections due to extended-spectrum beta-lactamase producing *enterobacteriaceae.* In only two of these eight studies, the patients had the same septic *focus:* Lower respiratory tract infection in one study (Rodríguez-Núñez et al.) and Osteomyelitis in the other (Gerlach et al.).

The quality of the eight studies included was assessed using the New Castle-Ottawa scale [8] and was moderate-low (Appendix A).

The four studies that evaluated all-cause mortality enrolled in total 148 patients: 87 patients treated with C/T alone and 61 patients treated with C/T association (Table 3). The clinical improvement outcome was evaluated in the majority of the studies, seven of eight (Table 4); enrolling in total 391 patients: 261 patients treated with C/T alone and 130 patients treated with C/T combination therapy. The two studies that evaluated the microbiological cure outcome enrolled a total of 33 patients: 13 patients treated with C/T alone and 20 patients treated with C/T combination therapy (Table 5).

### 3.2. Quantitative Synthesis

#### 3.2.1. All-Cause Mortality

The meta-analysis of the four studies evaluating all-cause mortality showed significant differences in C/T combination therapy compared to C/T monotherapy (*n* = 186 patients, OR: 0.31, 95% CI: 0.10–0.97, heterogeneity chi-squared *p* = 0.313, *p*-value = 0.045), the Forest plot is shown in Figure 2 All the studies reported an evaluation time of mortality at 30 days.

#### 3.2.2. Clinical Improvement

The meta-analysis of the seven studies reporting clinical improvement did not show significant differences in C/T combination therapy compared to C/T monotherapy (*n* = 432 patients, OR: 0.97, 95% CI: 0.54–1.74, heterogeneity chi-squared *p* = 0.954, *p*-value = 0.909), the Forest plot is shown in Figure 3. If the clinical improvement/cure was reported at different time points by the authors, we decided to analyze the longest follow-up but there was high heterogeneity in the time points of evaluation between the studies (Table 2).

#### 3.2.3. Microbiological Cure

The meta-analysis of the two studies that reported the microbiological cure did not show significant differences in C/T combination therapy compared to C/T monotherapy (*n* = 42 patients, OR: 0.83, 95% CI: 0.12–5.70), the Forest plot is shown in Figure 4.

## 4. Discussion

The main findings of this MA, based on the available evidence, were that there was a significant difference in the all-cause mortality outcome in favor of patients treated with C/T combination therapy compared to C/T monotherapy. These results were obtained by a low number of studies and patients (studies = 4, patients = 186). The data on clinical improvement showed no significant difference between C/T combination therapy compared to C/T monotherapy in a microbiological evaluable population. The overall effect on this outcome was evaluated from a relatively higher number of studies (*n* = 7) and patients (*n* = 391). For this reason, this finding should be considered the most robust of our analysis. The discrepancy between the overall estimate of effect between the all-cause mortality and clinical improvement outcome is difficult to interpret clinically, and should be investigated in future trials. However, the different sample sizes of the included patient cohorts and the different nature of the outcomes may explain the difference.

No difference in the clinical and microbiological improvement was observed in patients undergoing C/T combination therapy compared to C/T monotherapy for Gram-negative infections, in large part for the treatment of infections caused by *Pseudomonas aeruginosa*: 238 of the 391 patients included (60.8%). This finding could be useful in the optimization of the antibiotic treatment since the adequate knowledge of the new antibiotics will reduce their inappropriate use with the consequent reduction in the onset of new resistance and decreasing health care costs [17]. Regarding the microbiological cure, the very low number of included studies and patients preclude any meaningful interpretation.

Our results are not in line with a recent systematic review and network meta-analysis that compared an advanced-generation cephalosporin (ceftazidime) combined with a β-lactamase inhibitor (avibactam) [18]. The relatively low number of included patients in these meta-analyses suggest that further studies with the appropriate design should be conducted to evaluate the efficacy of combination therapy of newer antibiotics versus monotherapy.

Our results should be considered in light of some limitations. Our meta-analysis was based on data from a relatively low number of studies, of moderate-low quality, and low number of patients. Moreover, for the secondary outcomes we used the definitions provided by the authors. Therefore, the population may not be completely homogeneous. Other confounding factors could be that in only two studies there was a homogeneous population for focus of infection. Most studies (six out of eight studies) enroll patients with different types of infections. In addition, patients are not aligned with the organ failure rate and disease severity. Therefore, the confidence on the certainty of these results should be considered low. We used unadjusted data from the included nonrandomised studies, mostly retrospective and this approach may be biased by confounding. We did not evaluate adverse events as an outcome.

Unfortunately, the study of Rodriguez-Nunez 2019, enrolling ICU patients (MDR/XDR–LRI), that influenced the overall effect on the outcome of mortality at most, did not evaluate the microbiological and the clinical outcomes. It would have been interesting to investigate any discrepancies between these data.

In conclusion, the strength of this systematic review is the methodology that collected and synthetized all the available evidences with the suggestion that C/T combination therapy may reduce all-cause mortality compared to C/T monotherapy in infections due to Gram-negative bacteria but did not increase the rate of clinical improvement. However, the weaknesses of our meta-analysis is the low certainty of the evidence for these outcomes, limiting the impact of these findings. Further, clinical trials should evaluate the outcome mortality in order to give more objective and accurate information to clinicians.

## Figures and Tables

**Figure 1 antibiotics-10-00079-f001:**
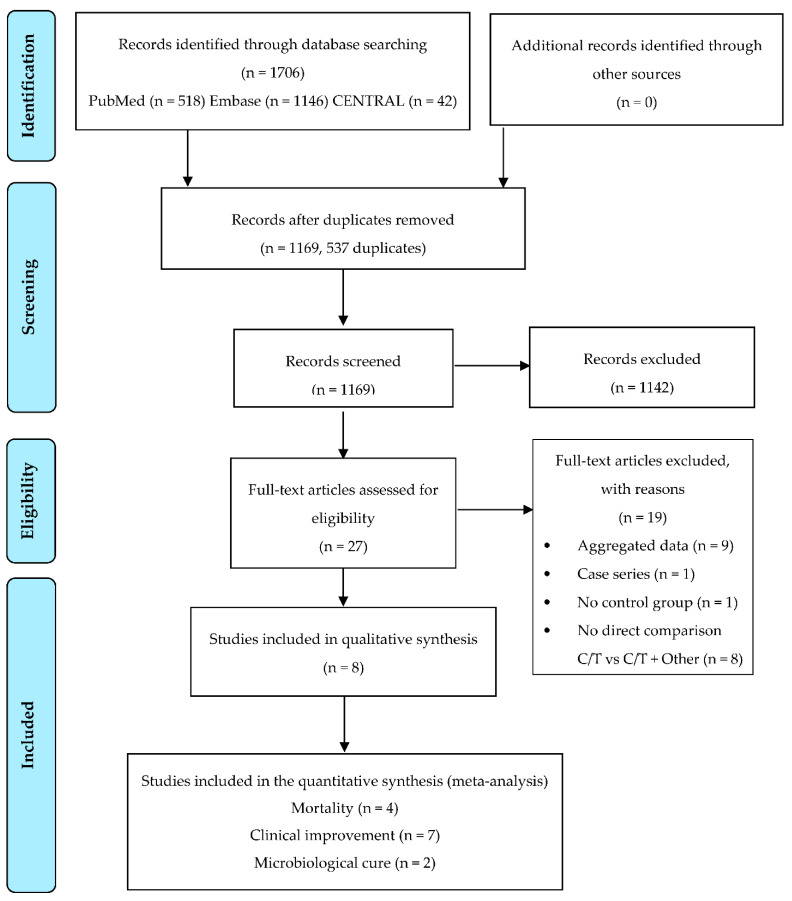
Flowchart of the study selection.

**Figure 2 antibiotics-10-00079-f002:**
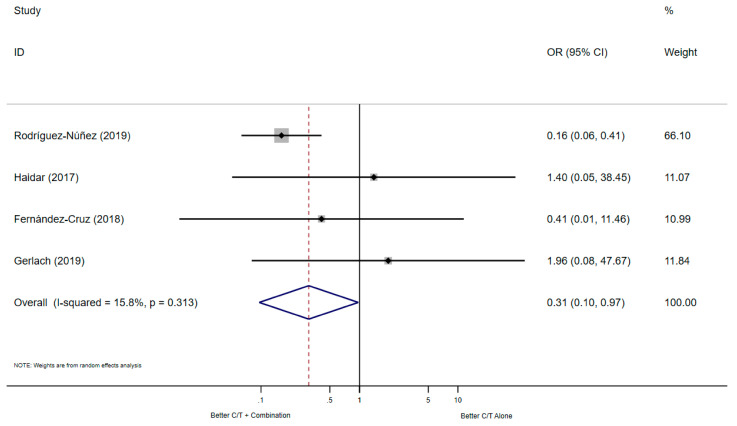
Forest plot of the four studies that reported the mortality as outcome.

**Figure 3 antibiotics-10-00079-f003:**
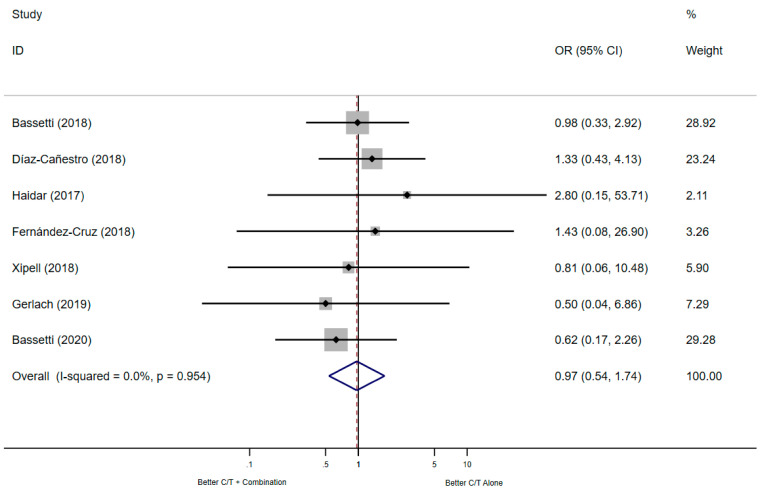
Forest plot of the seven studies that reported the non-clinical improvement.

**Figure 4 antibiotics-10-00079-f004:**
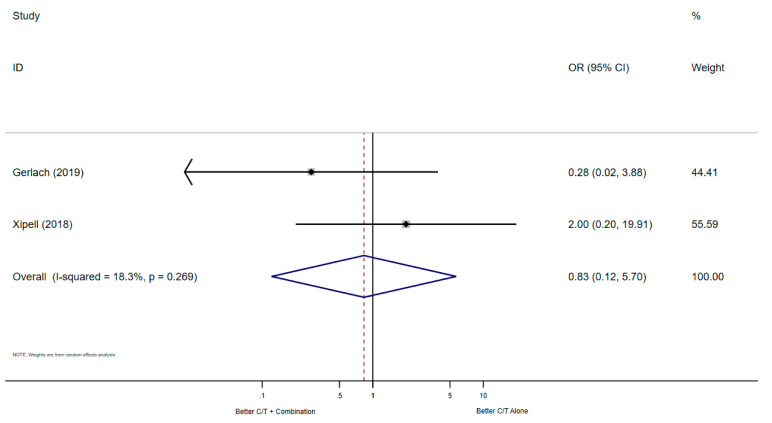
Forest plot of the seven studies that reported the microbiological cure.

**Table 1 antibiotics-10-00079-t001:** Population, Intervention, Comparison, and Outcome (PICO) method for selecting clinical studies in the systematic reviews.

Participants	Intervention	Comparison	Outcomes	Study Design
Adult patients in any setting with microbiological confirmed bacterial infection	Ceftolozane-tazobactam in association with another antibiotic/s	Ceftolozane-tazobactam alone	Primary outcomes: All-cause mortality Secondary outcomes: (a) Clinical improvement (b) Microbiological cure	Randomized controlled trials and observational Studies (including cohort and case–control studies)

**Table 2 antibiotics-10-00079-t002:** Summary of the studies included in the qualitative synthesis.

Author (Published Year) [Ref.]	Journal	Study Design	Country	Time Span	Pathogen	Septic Focus	Evaluation Time Points		
							Mortality	Clinical	Microbiological
Haidar (2017) [9]	Clinical Infectious Diseases	A single-center Retrospective cohort study	USA	06/15–03/16	MDR-PA	MIX	30-days	90-days	-
Fernández-Cruz (2018) [10]	Antimicrobial Agents and Chemotherapy	A single-center case-control study	Spain	03/16–02/18	PA	MIX	30-days	14-days	-
Xipell (2018) [11]	Journal of Global Antimicrobial Resistance	A single-center Retrospective cohort study	Spain	05/16–05/17	PA	MIX	-	NA	after 72 h of treatment
Bassetti (2018) [12]	International Journal of Antimicrobial Agents	Multicenter Retrospective cohort study	Italy	06/16–03/18	PA	MIX	-	MIX (7–23 M)	-
Díaz-Cañestro (2018) [13]	European Journal of Clinical Microbiology and Infectious Diseases	A single-center Retrospective cohort study	Spain	05/16–09/17	MDR/XDR-PA	MIX	-	after 7 days of treatment	-
Rodríguez-Núñez (2019) [14]	Open Forum Infectious Diseases	Multicentre Retrospective cohort study	USA France Spain UK	2016–2018	MDR/XDR-PA	LRI	30-days	-	-
Gerlach (2019) [15]	Infectious Diseases in Clinical Practice	A single-center Retrospective cohort study	USA	06/15–10/17	PA	Osteomyelitis	30-days	End-of-Therapy	any follow-up
Bassetti (2020) [16]	Open Forum Infectious Diseases	Multicentre Retrospective cohort study	Italy	06/2016–06/2019	ESBL	MIX	!	At the end of the follow-up period (August 2019)	-

! Clinical failure was defined as a composite of the following: (i) 30-day mortality; (ii) ongoing fever after 5 days of therapy; (iii) persistence of leukocytosis after 5 days of therapy; (iv) presence, after 5 days of therapy, of clinical signs of infection that could not be attributed to causes other than ESBL-E infection. PA: *Pseudomonas aeruginosa*; XDR: Multidrug-resistant; XDR: Extensively drug-resistant; ESBL: Extended-spectrum beta-lactamase; LRI: Lower respiratory tract infection; NA: Not available.

**Table 3 antibiotics-10-00079-t003:** Characteristics of the studies that evaluated the outcome mortality.

Author (Published Year) [Ref.]	Country	No. of Patients Enrolled	No. of Patients Treated with C/T Alone	No. of Patients Treated with C/T Association	No. of Patients Treated with BAT	BAT	C/T-Associated Antibiotic	Medical Ward
Haidar (2017) [9]	USA	21	19	2	X	X	+	NS
Fernández-Cruz (2018) [10]	Spain	57	11	8	38	¥	&	Hematological ward + Hematopoietic Stem Cell Transplantation Unit ICU: 12 (21.1%)
Rodríguez-Núñez (2019) [14]	USA France Spain UK	90	54	36	X	X	#	ICU (patients with LRI)
Gerlach (2019) [15]	USA	18	3	15	X	X	$	MIX ICU: 11 (61.1%)

#: Colistimethate, Aminoglycosides or Fluoroquinolones in 36 (40%); +: Ciprofloxacin 5 (23.8%), Tobramycin 2 (9.5%), Meropenem 1 (4.8%), Gentamicine 1 (4.8%), Imipenem 1 (4.8%); &: Levofloxacin 2 (3.5%), Amikacin 4 (22.1%), Colistin 1 (5.5%), or Fosfomycin 1 (5.5%); ¥: Piperacillin-Tazobactam, Cefepime, Ceftazidime, Meropenem, Ciprofloxacin, Colistin, or Amikacin as per in vitro susceptibility results; $: Ciprofloxacin 1 (5.5%), Daptomycin 8 (44.3%), Minocycline 4 (22.1%), Metronidazole 1 (5.5%), Polymyxin B 3 (16.7%), Trimethoprim/Sulfamethoxazole 1 (5.5%), Tobramycin 2 (11%), Vancomycin 2 (%). BAT: Best available therapy; NS: Not specified; ICU: Intensive care unit; LRI: Lower respiratory tract infection.

**Table 4 antibiotics-10-00079-t004:** Characteristics of the studies that evaluated the clinical outcome.

Author (Published Year) [Ref.]	Country	No. of Patients Enrolled	No. of Patients Treated with C/T Alone	No. of Patients Treated with C/T Association	No. of Patients Treated with BAT	BAT	C/T-Associated Antibiotic	Medical Ward
Haidar (2017) [9]	USA	21	19	2	X	X	+	NS
Díaz-Cañestro (2018) [13]	Spain	58	21	35	X	X	^^	MIX ICU: 16 (27.6%)
Bassetti (2018) [12]	Italy	101	65	36	X	X	£	MIX
Fernández-Cruz (2018) [10]	Spain	57	11	8	38	¥	&	Hematological ward + Hematopoietic Stem Cell Transplantation Unit ICU: 12 (21.1%)
Xipell (2018) [11]	Spain	24	15	9	X	X	<>	NS
Gerlach (2019) [15]	USA	18	3	14	X	X	$	MIX ICU: 11 (61.1%)
Bassetti (2020) [16]	Italy	153	127	26	X	X	NS	MIX ICU: 30 (19.6%)

BAT: Best available therapy; £: The most commonly used antibiotics were Aminoglycosides in 11 patients (10.9%), Colistin in 10 patients (9.9), and Carbapenems in five patients (5.0%); ^^ Mainly Colistin (45.9%), Amikacin (21.6%), Tobramycin (18.9%) + Ciprofloxacin 5 (23.8%), Tobramycin 2 (9.5%), Meropenem 1 (4.8%), Gentamicine 1 (4.8%), Imipenem 1 (4.8%) and Levofloxacin 2 (3.5%), Amikacin 4 (22.1%), Colistin 1 (5.5%), or Fosfomycin 1 (5.5%); ¥: Piperacillin-Tazobactam, Cefepime, Ceftazidime, Meropenem, Ciprofloxacin, Colistin, or Amikacin as per in vitro susceptibility results; <> Amikacin iv 7 (29.2%), Colistin iv 3 (12.5%), Tobramycin iv 1 (4.2%), Ciprofloxacin iv 1 (4.2%); $: Ciprofloxacin 1 (5.5%), Daptomycin 8 (44.3%), Minocycline 4 (22.1%), Metronidazole 1 (5.5%), Polymyxin B 3 (16.7%), Trimethoprim/Sulfamethoxazole 1 (5.5%), Tobramycin 2 (11%), Vancomycin 2 (%). NS: Not specified

**Table 5 antibiotics-10-00079-t005:** Characteristics of the studies that evaluated the microbiological outcome.

Author (Published Year) [Ref.]	Country	No. of Patients Enrolled	No. of Patients Treated with C/T Alone	No. of Patients Treated with C/T Association	No. of Patients Treated with BAT	BAT	C/T-Associated Antibiotic	Medical Ward
Xipell (2018) [11]	Spain	24	10	6	X	X	<>	NS
Gerlach (2019) [15]	USA	18	3	14	X	X	$	MIX ICU: 11 (61.1%)

BAT: Best available therapy; <>: Amikacin iv 7 (29.2%), Colistin iv 3 (12.5%), Tobramycin iv 1 (4.2%), Ciprofloxacin iv 1 (4.2%); $: Ciprofloxacin 1 (5.5%), Daptomycin 8 (44.3%), Minocycline 4 (22.1%), Metronidazole 1 (5.5%), Polymyxin B 3 (16.7%), Trimethoprim/Sulfamethoxazole 1 (5.5%), Tobramycin 2 (11%), Vancomycin 2 (%).

## Data Availability

The data that support the findings of this study are available on request from the corresponding author (M.F.).

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
