# Peer review of "Ceftolozane-Tazobactam Combination Therapy Compared to Ceftolozane-Tazobactam Monotherapy for the Treatment of Severe Infections: A Systematic Review and Meta-Analysis"

_antibiotics, 2021, doi:10.3390/antibiotics10010079_

Round 1
Reviewer 1 Report
This meta-analysis of Fiore et al. aims to compare C/T monotherapy to C/T-combination-therapy.
Abstract: line 35: “significant reduction” (p: 0.313) – which is the alpha-error-level?
Introduction: The introduction is very short and does not sufficiently lead to the topic and research question. In particular, the current level of knowledge on the topic would be interesting. Why is a meta-analysis on this topic needed?
Materials/Methods: Table 1 describes the procedure for ceftazidime/avibactam instead of ceftazidime/tazobactam. Please correct!
Line 95-96: Were the used definitions of secondary outcomes homogeneous?
Results: Line 130: this phrase is repeated
Overall, the result part would benefit from a bit more explanation. Currently, many facts are given mainly as references to tables/figures.
Table 4: what do you mean with “(not included in the meta analysis)” – this is misleading at this point.
Table 5: Why were those studies not included in the meta-analysis?
Figure 4: is there an error in the p-value again? 0.313 does not seem to be significant, but the 1 is not included in the confidence interval. Please clarify the alpha-level
Discussion:
Line 243-246: The lower mortality assessment was mainly driven by the study of Rodriguez-Nunez 2019 in ICU patients (MDR/XDR – LRI). Do you have an explanation for a better outcome in this group? Is there possibly a confounder? It would be interesting to get some more ideas why the discrepancy between outcome and microbiological cure was so high?
Please discuss the role of confounders in the discussion section in more detail.
Line 247: please clarify which difference is referred to here.
Please discuss the strengths and weaknesses of this meta-analysis in more detail in the context of available literature: How large and based on which quality of studies are comparable meta-analysis?
Line 261: Why should the reader still trust in your results?
Supplementary: Study [19] has four stars in outcome (only three stars possible). Please correct!
Author Response
Reviewer 1
This meta-analysis of Fiore et al. aims to compare C/T monotherapy to C/T-combination-therapy.
Abstract: line 35: “significant reduction” (p: 0.313) – which is the alpha-error-level?
- We apologize for the lack of clarity, the alpha-error-level is 0.05; p: 0.313 was related to heterogeneity, we corrected in p = 0.045 in the text (line 35) and p = 0.909 (line 39).
Introduction: The introduction is very short and does not sufficiently lead to the topic and research question. In particular, the current level of knowledge on the topic would be interesting. Why is a meta-analysis on this topic needed?
- We tried to clarify the usefulness, at this time, of a systematic review with meta-analysis (lines 56-60)
Materials/Methods: Table 1 describes the procedure for ceftazidime/avibactam instead of ceftazidime/tazobactam. Please correct!
- We apologize for the typo, we corrected it appropriately (line 78)
Line 95-96: Were the used definitions of secondary outcomes homogeneous?
- The definition of the secondary outcome “we used the definitions provided by the authors of included studies” (lines 99-100), we expressed this as a limit in the discussion (lines 316-317)
Results: Line 130: this phrase is repeated
- We apologize for the typo, we corrected it appropriately (line 147)
Overall, the result part would benefit from a bit more explanation. Currently, many facts are given mainly as references to tables/figures.
- We explained the results section in order to make it clearer (lines 123-127; 132-138)
Table 4: what do you mean with “(not included in the meta analysis)” – this is misleading at this point.
- We apologize for the typo, we corrected it appropriately (line 235)
Table 5: Why were those studies not included in the meta-analysis?
- We apologize for the typo, we corrected it appropriately (line 255)
Figure 4: is there an error in the p-value again? 0.313 does not seem to be significant, but the 1 is not included in the confidence interval. Please clarify the alpha-level
- we specified that p = 0.313 represented the heterogeneity chi-squared, the p-value was 0.045 (line 265)
Discussion:
Line 243-246: The lower mortality assessment was mainly driven by the study of Rodriguez-Nunez 2019 in ICU patients (MDR/XDR – LRI). Do you have an explanation for a better outcome in this group? Is there possibly a confounder? It would be interesting to get some more ideas why the discrepancy between outcome and microbiological cure was so high?
- Unfortunately, the study of Rodriguez-Nunez 2019, enrolling ICU patients (MDR/XDR – LRI), that influenced the overall effect on the outcome of mortality at most, did not evaluate the microbiological and the clinical outcomes. It would have been interesting to investigate any discrepancies between these data. (lines 324-327)
Please discuss the role of confounders in the discussion section in more detail.
- We discussed the role of confounders in the discussion (lines 316-320)
Line 247: please clarify which difference is referred to here.
- We clarified that “No difference in clinical and microbiological improvement was observed” (line 301)
Please discuss the strengths and weaknesses of this meta-analysis in more detail in the context of available literature: How large and based on which quality of studies are comparable meta-analysis?
- We highlighted the strengths and weaknesses of this meta-analysis as suggested (lines 328-332)
Line 261: Why should the reader still trust in your results?
- The low certainty of the evidence that we reported should be linked to the low quality of the available evidence, rather than to our review design or methods.
We believe that our systematic review and meta-analysis could act as a call for better research on this topic. However, we also believe that our study has importance at this stage to inform the readers about the quality of evidence regarding the effectiveness of combination therapy with ceftolozane/tazobactam vs. monotherapy for the treatment of severe infections. We have now extensively discussed the interpretation of our findings in light of the quality of available evidence.
Supplementary: Study [19] has four stars in outcome (only three stars possible). Please correct!
- We apologize for the typo, we corrected it appropriately
Reviewer 2 Report
This systematic reviwe and meta analysis written by Marco Fiore et al. is very interesting and answer an important question for clinical management of patients infected with XDR.
Some comments need to be addressed.
Lines 80 to 89 could be fuse into one unique paragraph.
Line 130 is a duplication of the line 129.
Figure 1 : indicate how many additional records identified through other sources, and the causes of exclusion for the 1362 references.
Tables 2 to 5 : please indicate the national location of the described studies, as the recommendations could vary between countries.
Discussion : Paragraph lines259-263 : authors have to justify why they have not addressed the limits they describe, or address it.
Author Response
Reviewer 2
This systematic review and meta analysis written by Marco Fiore et al. is very interesting and answer an important question for clinical management of patients infected with XDR.
Some comments need to be addressed.
Lines 80 to 89 could be fuse into one unique paragraph.
- Many thanks, we fused the lines into one unique paragraph as suggested (lines 81-93)
Line 130 is a duplication of the line 129.
- We apologize for the typo, we corrected it appropriately (line 147)
Figure 1 : indicate how many additional records identified through other sources, and the causes of exclusion for the 1362 references.
- We indicated in the Figure how may additional records we identified through other sources (line 170) and added in the results the causes of exclusion for the records excluded (lines 123-127). Furthermore, we added a Supplementary Table (Table S1), with all the paper excluded ad the reason of exclusion.
Tables 2 to 5 : please indicate the national location of the described studies, as the recommendations could vary between countries.
- We indicated the national location as requested
Discussion : Paragraph lines259-263 : authors have to justify why they have not addressed the limits they describe, or address it.
- We discussed more extensively the limits of our findings highlighting the strengths and weaknesses of this meta-analysis as suggested (lines 314-334)
Round 2
Reviewer 1 Report
Thank you for majorly improving your manuscript.